# Piezo1 in Digestive System Function and Dysfunction

**DOI:** 10.3390/ijms241612953

**Published:** 2023-08-19

**Authors:** Jing He, Xiaotian Xie, Zhuanglong Xiao, Wei Qian, Lei Zhang, Xiaohua Hou

**Affiliations:** Department of Gastroenterology, Union Hospital, Tongji Medical College, Huazhong University of Science and Technology, Wuhan 430022, China; hejing199909@163.com (J.H.); 18883147537@163.com (X.X.); xzlcns1002@126.com (Z.X.); qianwei63@126.com (W.Q.)

**Keywords:** mechanotransduction, Piezo1, mechanosensitive ion channel, digestive system, biological function

## Abstract

Piezo1, a non-selective cation channel directly activated by mechanical forces, is widely expressed in the digestive system and participates in biological functions physiologically and pathologically. In this review, we summarized the latest insights on Piezo1’s cellular effect across the entire digestive system, and discussed the role of Piezo1 in various aspects including ingestion and digestion, material metabolism, enteric nervous system, intestinal barrier, and inflammatory response within digestive system. The goal of this comprehensive review is to provide a solid foundation for future research about Piezo1 in digestive system physiologically and pathologically.

## 1. Mechanosensitive Ion Channels in the Digestive System

Mechanotransduction refers to the process of living mechanosensitive tissues or cells to detect and respond to changes in membrane tension and cytoskeleton induced by mechanical stimuli, initiating intracellular signal transduction and generating electrochemical signals [1,2]. The digestive system experiences various mechanical stimuli, including gastrointestinal peristalsis, villus movement, conduit osmotic pressure, etc, which are fundamental for initiating mechanotransduction. Mechanotransduction relies on ion channels sensitive to mechanical stimuli, which are known as mechanosensitive ion channels. The mechanosensitive ion channels in digestive system include transient receptor potential vanilloid family (TRPV) [3], Piezo1/2 [4], two pore-domain potassium channels (K2p) [5], large-conductance Ca^2+^-activated potassium channel (BKCa) [6] and others.

The Piezo protein, characterized as the largest plasma membrane ion channel complex with over 30 putative transmembrane domains, is a unique entity capable of inducing large mechanically-activated cationic currents unlike other known ion channels or proteins [7]. At present, Piezo channel has drawn considerable research interest [8]. Piezo protein contains two homologues in *Homo sapiens*, Piezo1 and Piezo2 (Table 1). Compared with Piezo1, Piezo2 has additional charged residues at the beam-carboxy-terminal domain interface and additional constriction sites at L2743, F2754 and E2757 in the central pore [9,10]. Piezo1 is widely expressed in multiple cell types, whereas Piezo2 is believed to be predominantly expressed in neurons and intestinal enterochromaffin cells.

First identified in 2010 by Coste B [11], Piezo1 (Fam38a) plays important roles in maintaining various cellular effects such as bone and epithelial homeostasis, neural stem cell differentiation, macrophage polarization, and regulating biological functions including vascular development, red blood cell volume homeostasis, inflammation response generation and etc. [12,13,14,15]. More recently, Piezo1 channel has also been identified to transduce itch in sensory neuron which is associated with Piezo2 channel as generally believed [16]. Therefore, Piezo1 participates in life activity widely and deeply. And more and more evidences have demonstrated the predominant and special contributions of the Piezo1 channel in the digestive system at present [4]. here we review current studies focused on the cellular effects of Piezo1 in digestive system, with special highlights on its importance in regulating biological function.

**Table 1 ijms-24-12953-t001:** Differences between Piezo1 and Piezo2.

	Piezo1	Study	Piezo2	Study
Gene region	16q24.3	[12]	18p11.22-p11.21	[17]
Amino acid residues	2520	[12]	2752	[17]
Distribution	almost all cell types	-	mainly nerve cells and some specific cell types	[18]
Detection threshold (fJ) *	213.7 ± 16.6	[19]	86.8 ± 7.1	[19]
Work resolution (fJ) *	1.2 ± 0.4	[19]	1.0 ± 0.2	[19]
Transduction Speed (ms) *	8.2 ± 2.2	[19]	1.5 ± 0.5	[19]
Inactivation kinetics (ms) *	16.5 ± 1.4	[11]	7.3 ± 0.7	[11]
Activator	Yoda1, Jedi1/2	[20,21]	-	-
Inhibitor	Dooku1, GsMTx4, ruthenium red (RR), gadolinium (Gd^3+^)	[11,22]	GsMTx4, ruthenium red, gadolinium	[23]
Hereditary human disorders	dehydrated hereditary stomatocytosis, congenital lymphatic dysplasia with non-immune fetal hydrops	[24,25,26]	recessive distal arthrogryposis syndrome, dominant distal arthrogryposis syndrome (type III and V), Marden-Walker Syndrome	[27,28]

* detected in HEK293T cell line.

## 2. Structure and Kinetics of Piezo1

The Piezo1 gene is located on human chromosome 16q24.3 and contains 51 exons [12]. The Piezo1 monomer is about 290–320 kDa while it is a conservative trimer of about 900 kDa naturally [29]. This structure mediats nonlinear transduction of mechanical energy and detects mechanical energies as 213.7 fJ, with a resolution of 1.2 fJ [19]. As a non-selective mechanosensitive cation channel, Piezo1 has the strongest affinity to Ca^2+^ [11,30]. The opening of Piezo channel triggers mainly influx of Ca^2+^ and Na^+^ [31,32]. Therefore, opening of Piezo channel has two effects: it shifts membrane potential and activates other voltage-gated ion channels, which triggers an action potential; simultaneously, it alters [Ca^2+^]_i_ (intracellular calcium) concentration and triggers downstream signal transduction pathways [33]. Structurally, Piezo1 can be divided into three parts: peripheral N-terminal propeller blades for sensing mechanical stimulation, beam and anchor domain for conducting mechanical signals, and the C-terminal central pore for facilitating ion transport [12] (Figure 1).

Currently, two principal mechanisms are proposed for Piezo1 activation: the “force-from-lipids” mechanism believes mechanical force alters the membrane lipid-Piezo1 interaction and induces the activation of Piezo1; the “force-from-filaments “ mechanism suggests the force modifies the interaction between Piezo1 and extracellular matrix or cytoskeletal proteins, thereby changing their conformation and opening the channels [34,35,36]. The current induced by channel opening gradually weakens and deactivates slowly at positive membrane potential but does so rapidly at negative membrane potential [11], which may be associated to the extracellular domain and inner helix in the central pore [37,38,39].

## 3. The Main Cellular Effects of Piezo1 in Digestive System

First, we have reviewed the role of Piezo1 of digestive system at the cellular level, to summarize the main cellular effects of Piezo1. The overall cellular effects are list in Table 2, and the associated signal network are shown in Figure 2.

### 3.1. Basic Cell Activities

The fundamental life activity of cell includes cell proliferation, differentiation, fertilization, senescence and death, etc. Recent perspectives suggest mechanical cues significantly regulate these processed [80,81], which may be medicated by Piezo1 in the digestive system.

It is generally believed that Piezo1 promotes cell proliferation and migration, and inhibits cell apoptosis. Study showed that Piezo1 activates the TGF-β pathway to induce cell proliferation and epithelial-mesenchymal transition in hepatoma cell [69]. The similar phenomenon has been observed in periodontal ligament stem cell: activation of Piezo1 significantly accelerates periodontal tissue growth via the induction of Leptin receptor+ cells [82]. Specific mechanisms includes Piezo1-induced cell proliferation via MAPK-ERK1/2 pathway and purinergic signaling transduction [43,44], and cell migration promotion through induces HIF-1α/VEGF expression (may be related with adaption to hypoxia environment) [53,70,71] and regulates integrin expression (may be related to cell adhesion) [83]. However, the cellular effects of Piezo1 activation is complicated, as it not only promotes cell migration but also mediates apoptosis by decreasing mitochondrial membrane potential [64], although someone emphasized the need of cytotoxic Ca^2+^_[i]_ levels mediated by extensive activation of Piezo1 in the apoptosis process [84]. This dual role of Piezo1 activation may correlate with the dose-response curve of Yoda1 administration [85,86]. Moreover, one potential apoptosis-associated mechanism is that Piezo1 activation induces the extrusion of epithelial cells, different from the pro-proliferation role of Piezo1 as well [87,88]. Given that Piezo mechanosensitivity is multifactorial [89], precise cellular effects of Piezo1 requires accurate intervention such as optogenetics to be clarified in specific conditions [90].

Piezo1 is involved in cell differentiation and development in digestive system. Mechanical cues direct intestinal stem cells to differentiate into goblet cells in IBD [91]. Similarly, the mechanosensitive Piezo expressed in fly intestinal stem cells mediates the differentiation to secretory enteroendocrine cells [57]. Intestinal organoids provide a reliable in vitro model of intestinal stem cells [92]. Another study refined that the Piezo1 component is abundant in inflated intestinal organoids, regulating stem cell fission and differentiation associated with stretch state [93]. This cellular effect of Piezo1 on acinar cell differentiation even influences organ morphogenesis in submandibular gland [46]. Cell differentiation associated downstream responses induced by Piezo1 activation including integrin, ERK1/2-MAPK, Notch, and WNT signal pathway as reported [94].

### 3.2. Immune Signal Transduction—Initiation, Recruitment, and Diapedesis

Traditionally, intestinal infection occurs due to pathogenic bacteria through occupying biological niches [95], producing immunogenic substances [96] and secreting pathogenic factors [97]. Emerging evidence indicates that biophysical cues may also activate immune cell and immune responses, in which Piezo1 stimulates innate immune cells to elicit an inflammatory response as a mechanotransduction role [98,99]. During intestinal infection, pathogenic bacteria activate Piezo1 through invasion-induced epithelium membrane ruffles, triggering ATP secretion and evoking gene expression in immune and barrier pathways [55]. Intriguingly, Piezo1 activation induced by Granzyme A in colon epithelial cells of children with IBD activates cell autophagy via phosphorylating NF-κB p65. [62]. This result imply that intestinal immunity response affects mechanotransduction partially through membrane perforation and Piezo1 mechanosensation.

Peyer’s patch is an important structure of immune response in the small intestine and its conduit network is formed by collagen fibers and fibroblast reticular cell, which transport fluid and immune cells to the lymph node parenchyma [58,100,101]. Piezo1 is expressed in fibroblast reticular cell and responds to fluid flow in the conduits. Dysfunction of Piezo1 damages lymphocyte recruitment homeostasis of Peyer’s patches and inhibits the mucosal immunity [58].

Recruited immune cells pass through blood vessels after transform, migration, and diapedesis process, etc. Here, cellular effect of Piezo1 in vascular endothelial cells plays an essential role in initiating leukocyte diapedesis. Piezo1-deletion in endothelial cells significantly reduces the number of CD11b/Ly6G myeloid cells in the peritoneal cavity in peritonitis model [102].

### 3.3. Cell-Specific Activities—Sensory Transduction, Antioxidation

As a mechanosensitive ion channel, Piezo1 plays a pivotal role in mediating mechanical sensation and sensory conduction. The trigeminal nerve dominates the sensation of head, face, and oral cavity. Piezo1 of odontoblasts cell mediates ATP secretion, contributing to dentinal sensitivity generation and sensory conduction in trigeminal ganglion neurons [40,41]. Notably, Piezo1 is also present in the myelinated axon of trigeminal ganglion neurons projected to dental pulp, where it is involved in axonal plasma membrane disappearance and mediates acute pain [42]. Sensory neurons in the vagus nerve, which form part of the “gut-to-brain axis”, detect stretch and/or nutrients signal in the digestive system. While GLP1R neurons senses mechanical distension of the stomach and intestine [103], it is also reported that Piezo1 exists in the vagus nerves [104] but we don’t know if Piezo1 mediates associated sensory conduction.

Generally, activation of Piezo1 triggers Ca^2+^ influx, leading to reactive oxygen species generation when Ca^2+^ overload occurs. Contrary to hypothetical cellular effect of Piezo1 activation, antioxidation induced by Piezo1 activation has also been discovered in hepatocytes. Nrf2 is an important antioxidant and anti-inflammatory regulator promoting the expression of glutathione-S-transferases (Gst) and NAD(P)H: quinone oxidoreductase 1 (Nqo1) [105,106]. Piezo1 activation in hepatocyte increases the level of Nrf2 and Nqo1/Gsta1 genes to reduce cell death and mitochondrial oxidative stress [67]. This antioxidative response may be linked to the increase in Ca^2+^/CaMKII seen under stress conditions after TRPV1 channel activation [107]. Another point worthy to mention is that redox regulates calcium ion channels although research on their specific interaction with Piezo1 is currently absent [108].

## 4. Piezo1 Affects the Biological Function of the Digestive System

Next, we analysis the role of Piezo1 in the biological function of the digestive system (Figure 3) at physiological and pathological perspective, to obtain a more comprehensive cognition.

### 4.1. Ingestion and Digestion

Generally speaking, food ingestion generates mechanical distension and triggers gastrointestinal hormone secretion, which evokes gut-brain communication via vagal afferents/blood stream to regulate eating behavior and gastrointestinal motor functions [109,110]. It is postulated that Piezo1 plays a part in this feedback loop. Piezo-expressed diuretic hormone 44 (DH44) neurons project to the Crop (analogous to the human stomach) [111] and are activated by ingestion stretch and food flow when the Crop distends beyond 0.2 μL [112]. This activation results in suppressed DH44-positive neuron activity, reduced sugar intake, decreased excretion, and is associated with food choice behavior [111]. Consistently, activation of Piezo1 in pharynx and anterior gut by food intake weakens pharyngeal pumping, pharyngeal gland activation and decreases appetite [48,113]. In fact, the disorder of mechanical signal such as delayed gastric emptying is associated with purging behaviors in clinical experiment [114]. We don’t know if Piezo mediated this reaction. However, it is believed that it is Piezo2 be related to ingestion in mammalian, while there still lack of evidence on the relationship between Piezo1 and ingestion in human at present.

The digestion is regulated by mechanosensitive ion channel [115], and Piezo1 deeply participates in and promotes the secretion of digestive hormones and juice. Gastrin, a vital hormone regulating stomach activity, is predominantly secreted in response to protein decomposition products and gastric wall distension [116]. Piezo1 is concentrated at the base of G cells in gastric antrum, enabling G cells to secret gastrin directly in response to antral expansion [50]. Bile synthesis is determined by a “mechano-osmotic” process, and bile flow is facilitated by Ca^2+^-stimulated contractions in the peritubular actin cortex [117]. Human liver produces bile from the rate of 0.5–1 mL/min during fasting to 2–3 mL/min after feeding [118], resulting in sharply increasing biliary pressure. A study has reported that Piezo1 acts as the tension sensor in bile canaliculi membrane, and is activated to provoke the contraction of peritubular actin cortex, thereby propelling bile acid into the intrahepatic bile duct from hepatocyte and bile canaliculus [74]. When the bile acid flows to intrahepatic bile duct, hypotonic stress of cholangiocytes during cholestasis activates Piezo1 and further promotes the secretion of bile acid and bicarbonate via Pannexin1-ATP-P2X4R axis [75]. Besides cAMP-activated CFTR channel, calcium-mediated pathways appear to dominate bile secretion initiated by secretin; on the other hand, Piezo1-activated ATP release may clarify the unknown mechanism of ATP transport into the lumen [117]. Therefore, Piezo1 definitely has a profound impact on the research in bile secretion and transport. Moreover, Piezo1 expression has been detected in all islet cells, and Piezo1 activation in β cells induces insulin secretion in hypotonic conditions following glucose absorption [77]. Therefore, glucose triggers insulin secretion not only through classical membrane depolarization following the closure of K_ATP_ channels and voltage-gated Ca^2+^ channels [119], but also through a Piezo1-involved mechanotransduction pathway.

The rectum and anus are sensitive to mechanical stimuli [120,121], but evidence supporting Piezo1’s role in the mammalian defecation system is scarce so far. A study showed that the mechanotransduction of anus sensory neuron relies on the TRP channel NOMPC but not Piezo in defecation circuitry [122]. However, a study argued that Piezo1 is expressed in anal cells, knockout of which resulted in shorter defecation periods and increased defecation frequency [48]. Generally speaking, we can’t deny the role of mechanosensitive Piezo in defecation circuitry because the defecation system requires the coordinated actions among intestine, muscle, nerve, etc, and the sensory receptors are also present in the extrarectal tissues and pelvic floor [123]. In fact, Piezo2 play a role in sensing the luminal forces and luminal contents to regulate transit times in the intestine [124]. Further researches are needed to determine the exact role of Piezo1 in human rectum and anus.

### 4.2. Material Transport and Metabolism

The intestine serves as a vital organ for absorbing external substances and transporting them to circulation via intestinal epithelium [125]. It seems that Piezo1 is a significant player for epithelial transport. The divalent metal transporter 1 (DMT1) is an essential component in the entry of mercury into the intestinal epithelium [126]. Fluid flow and cyclic mechanical stretching mimicking the mechanical environment within the intestine upregulates Piezo1 in intestinal epithelial cell and boosts DMT1 expression in the golgi via calpain activation, corresponding with a rise in the absorption of nonbiodegradable mercury ions [63]. In fact, a study has showed a similar interaction between mechanical stimulation and epithelial absorption in a fish model [127]. Another interesting point is that Piezo1 expression in colon cancer is increased with lovastatin and decreased with water-soluble cholesterol MβCD-CHOL [128]. This finding implies that epithelial dietary transport affects intestinal mechanotransduction by regulating cellular cholesterol homeostasis and Piezo1 expression in the large intestine, further indicating the possibility of “force-from-lipids” mechanism of Piezo1 activation.

Liver is the central conductor of systemic iron balance [129], and macrophages in the spleen and liver degrade senescent/damaged erythrocytes through phagocytosis and export iron from heme [130]. This process generates several mechanical cues, including the interaction of macrophages with red blood cells and macrophages movement/deformation. Activated Piezo1 of liver macrophages enhances phagocytic activity and accelerates erythrocyte turnover, leading to increased iron release and body iron overload [72]. On the other hand, gain-of-function Piezo1 in hepatocytes inhibits the expression of hepcidin, a hepatic iron regulator hormone, by inhibiting the transcription of HAMP gene and impairs liver iron metabolism [68]. Therefore, a mild gain-of-function Piezo1 allele, E756del is considered as a risk factor for iron overload [72]. Symptoms such as low hepcidin and iron overload in patients with hereditary stomatocytosis have clinically corroborated that Piezo1 may disrupt liver iron metabolism [131,132].

One attractive field is that Piezo1 may be a potential target linking mechanotransduction and glycolysis reported in colon macrophages. Macrophages are recruited to inflamed tissue during IBD and secrete numerous inflammatory factors such as IL-1β, IL-6, and TNF-α [133]. Altered microenvironmental stiffness during inflammation is sensed by Piezo1 in macrophages [14]. The activated Piezo1 promotes the shift of macrophages metabolism into aerobic glycolysis and increases the secretion of IL-1β, IL-6, TNF-α, while knock-out of Piezo1 inhibits aerobic glycolysis in colon macrophages of colitis induced by dextran sulfate sodium salt [65]. Similarly, Piezo1 reportedly enhances anaerobic glycolysis and mitochondrial respiration in vascular endothelial cell to stimulate ATP production [134], further indicating a significant association between mechanosensitive Piezo1 and glucose metabolism.

### 4.3. Enteric Nervous System and Gastrointestinal Motility

The enteric nervous system (ENS) is the primary commander for gastrointestinal motility with certain myenteric plexus neurons responding to mechanical cues to facilitate motor function [135]. In the enteric nervous system, Piezo1 is expressed in 50–80% submucosal plexus which co-expresses vasoactive intestinal peptide, and in 15–35% myenteric plexus which mainly co-expresses nitric oxide synthase [51]. There was a positive correlation between gastrointestinal mechanosensitive neurons and Piezo1-positive neurons, but the latter do not significantly contribute to the mechanosensitivity of gastrointestinal plexus [51]. Another study showed that distension in distal colon induced Ca^2+^ elevations in neurons and regulated the excitability of ENS circuits via mechanosensitive channel, unaffected by GsMTx4 but suppressed by KCa1.1 channel inhibition [136]. Thus, the role of Piezo1 in direct ENS mechanotransduction should be interpreted cautiously. By comparison, many studies highlight the robust mechanosensitive role of Piezo2 in ENS [137,138,139]. We speculate the difference in mechanotransduction between Piezo1 and Piezo2 originates from the kinetics differences despite their similar structure. With its higher sensitivity and faster signal transduction and inactivation (Table 1), Piezo2 appears better suited to rapid response needed in nervous system. Notably, the Piezo1 activity varies in different nerves even with standardized expression [140]. Considering Piezo1 definitely play a mechnosensitive role in nerves [141], we also have to pay attention to the difference between ENS and other nervous system, which are different in development, differentiation, and function [142,143,144].

The interaction between ENS and the immune cells, gut microbiota, and enteroendocrine cells is associated with the organization and function of enteric motor circuits [145]. Some studies indicate Piezo1 participates in this interaction. 5-HT, for instance, plays a significant role in stimulation of propulsive and segmentation motility patterns, epithelial secretion, vasodilation, inflammation and serves as a trophic factor in the intestine [146,147]. Intraperitoneal injection of Yoda1 alleviated intestinal propulsion and contraction disorders in water avoidance stress mice [60], suggesting a possible interaction between Piezo1 and gastrointestinal motility [148]. In fact, a role of Piezo1 in mechanotransduction of gastric smooth muscle cells has been suggested [149]. Generally, Piezo1 is a positive regulator of gastrointestinal movement and also mediates the synthesis of 5-HT.

The pattern of Piezo promotes intestinal 5-HT synthesis is special: Piezo1 is evoked by intestinal microbiota ssRNA to regulate the expression of Trp1, while mechanical stimulation boosts intestinal 5-HT synthesis in a Piezo2-dependent manner [23,54,150]. Therefore, we have to consider the non-mechanosensitive role of Piezo1. Considering that enterochromaffin cells contribute to approximately 95% of 5-HT production in human [149], we postulate that both the microbiome-ssRNA-Piezo1 axis and the mechanical stimulation-Piezo2 axis affect systemic 5-HT synthesis level and promote the local intestinal movement (Figure 4). Nevertheless, it should be noted that Alec R Nickolls reported that ssRNA stimulates calcium influx in certain cell lines, but this response is independent of Piezo1 [89]. Study reported RNA binding to zwitterionic lipid bilayers in the presence of divalent cations and ssRNA has a strong preference to reside in isotropic solution rather than in association with an inverse hexagonal phase of a lipid [151]. However, intestinal fluid contains many divalent cations and we can’t dismiss the “force-from-lipids” or “force-from-filaments” mechanism of Piezo1 activation induced by ssRNA. At present, there exists too scarce data to diagnose this “dilemma disease”, and further studies are needed to determine whether Piezo1 serves as the receptor for ssRNA in the gut.

### 4.4. Intestinal Barrier

Generally, the intestinal barrier includes mucus barrier, epithelial barrier, endothelial barrier and biological barrier, all of which are exposed to diverse mechanical stimulation. In brief, Piezo1 is expressed throughout the intestinal epithelium and contributes to the epithelial barrier through acting on cell extrusion and tight junction. The integrity of intestinal epithelial barrier depends on stable cell turnover and constant cell numbers [152]. When epithelial cells crowds and cytoskeleton contracts, Piezo1 is activated to promote the apical extrusion; while mechanosensitive Piezo1 disruption inhibits extrusion and forms epithelial cell masses [87,88], resulting in imbalanced cell numbers and impaired epithelial barrier. Tight junction seals the epithelial and endothelial monolayers, and it is formed by associated protein such as claudins, occludin, ZO-1. Study reported Piezo1 regulates the expression of claudin-1 via ROCK1/2 signal [61]. Overexpressed Piezo1 downregulates claudin-1 expression and increases intestinal epithelial permeability. Interestingly, cell-cell junction itself is mechanosensitive [153] and a recent study showed that membrane Piezo1 interact with CD31 and is dragged to the tight junction of endothelium [154]. Perhaps there exists a unique synergy mechanism for mechanotransduction between the two adjacent complexes.

Similarly, Piezo1 activation in the intestinal vascular endothelium may disrupt endothelial barrier but direct evidence is lacked at present. In HUVEC study, Piezo1 accumulates at the leading apical lamellipodia under shear stress and rearrangement endothelial cells [66]. Yoda1 treatment in HUVEC also leads to cell-cell junction disruption, radial actin collapse [154], and downregulated VE-cadherin expression [85]. These effects increase vascular permeability and disturb vascular barrier in vitro, but intestinal evidences in vivo are required.

Piezo1 in goblet cell is essential for maintaining the colon mucous barrier and intestinal microflora balance. Goblet cells synthesize and secrete mucin, forming the mucus barrier in the gastrointestinal tract [155]. Piezo1 is highly expressed in goblet cells and functions as the mechanical sensor and mucin2 regulator [59,60]. Knockout of Piezo1 in goblet cells results in decreased mucin2 expression, goblet cell numbers, mucus layer thickness, and increased inflammatory cytokines [156]. For a long time, researchers have found mechanical stimulation increase the number and activity of goblet cells [59,157]. Piezo1 gives a potential interpretation for this relationship since Piezo1 is the key mechanical sensor in intestinal epithelium compared with TRPA1, TRPV1, TRPV4, TRPV6, TLR4 and ASIC [59].

Intestinal microflora is an integral part of biological barrier of intestine [158]. It is worth mentioning that there also exist mechanosensitive channels in intestinal microflora, such as MscL and MscS expressed on *Escherichia coli* [159]. However, whether Piezo1 is directly expressed and its role in intestinal flora remain to be studied.

### 4.5. Inflammatory Response

Generally, inflammation is initiated by infection, tissue injury, tissue stress and malfunction, causing a range of complications including autoimmunity, tissue damage, sepsis, and fibrosis [160]. During inflammation, tissue undergoes edema in the acute phase and stiff fibrosis in the chronic phase, both of which provide mechanical cues for mechanotransduction. In brief, it has been observed that Piezo1 plays a critical pro-inflammatory role in every stage of the inflammatory response within the digestive system.

Infection is the most common initiations in inflammation. As mentioned above, Piezo1 is activated by invasion-induced epithelium membrane ruffles of pathogenic bacteria to initiate immune signaling upon infection [55]. However, Piezo1 deletion in goblet cells increases the diversity and abundance of mucosa-associated microorganisms such as *Helicobacter hepaticus*, *Lactobacillus johnsonii*, colibacin-producing *Escherichia-Shigella* and *Oscillospiracea* [156]. It seems that mechanotransduction interacts with intestinal bacterial infection in a complicated circuit. We are cautious about the “anti-inflammation” role in this condition. As a mechanosensitive ion channel physiological-required, Piezo1 should be research under mechanical stimulation or disease condition. Indeed, another study in DSS model showed that gut-specific Piezo1 deficiency led to minimal microbiome abnormalities [54]. And Piezo1 in intestinal goblet cell has already been identified as an IBD risk gene by a genome-wide association study [161].

Inflammatory cells extravasate to nidus from vessel during inflammation. As observed, the intestinal focus of IBD patients undergoes pathological vascular remodeling and angiogenesis [162]. However, the new vessels are immature, insufficiently perfused, leaky, and hypersensitive to growth factors, thereby promoting inflammatory cell recruitment and IBD progression [163,164]. Piezo1 is expressed in colon microvascular endothelial cells [66] and is activated by stimulation like blood flow disorder during colitis [165]. Knockout of Piezo1 in endothelial cells inhibits formation of lumen induced by shear stress and significantly reduces the number of invading endothelial cells, thickness of sprouts, and invasion distance of sprouts [166]. Furthermore, attenuated intestinal glycolysis through PFKFB3 inhibition reduced pathological angiogenesis in DSS-induced colitis [167,168]. A study reported activation of Piezo1 by Yoda1 enhanced mitochondrial respiration and glycolysis in HUVEC [134]. Given that intestinal inflammation leads to tissue remodeling [169] and alters the mechanical environment of endothelial cells, Piezo1 may contribute to intestinal nutrition supply and inflammation by promoting vascular remodeling and angiogenesis.

Moreover, Piezo1 participates in the production and secretion of inflammatory mediators in the mechanotransduction process, which in turn alter the function of tissues and organs. IL-6, TNF-α, and IL-1β are common and important inflammatory mediators involved in intestinal acute phase inflammation. Piezo1 activation stimulates their production and secretion in intestinal recruited macrophages and intestinal epithelial cells during IBD [56,65]. Gut-specific Piezo1 deletion significantly downregulates the expression of pro-inflammatory cytokines (TNF-α, IL-17, and IL-5) [54], highlighting Piezo1 as a potential prophylactic target for IBD. Another inflammatory mediator, CXCL1 is also reported to be triggered by Piezo1 of liver sinusoidal endothelial cell under hepatic congestion and cyclic stretch condition [73]. As one of the most critical chemokines in inflammation [170], CXCL1 recruits neutrophils in hepatic sinusoid, inducing microthrombosis and increasing portal vein pressure [73]. This function of Piezo1 illustrates partly a role in the initiation of increased intrahepatic resistance and consequent portal hypertension [171].

Inflammation could also be induced endogenously by stressed, or dead cells [160]. When pancreas pressure increases, Piezo1 in the pancreatic acinar and stellate cells is activated and triggers the opening of TRPV4 channel consequently. This action leads to a sustained increase in [Ca^2+^]_i_, and Piezo1-initiated [Ca^2+^]_i_ overload causes pancreatitis through pancreatic acinar cell necrosis/trypsin activation in the acute phase, and fibrosis through increased TGF-β1/fibronectin/collagen I expression in the chronic phase [78,79,172]. Interestingly, the interplay between Piezo1 and TRPV4, both mechanosensitive ion channels, appears to be complex and extends beyond the pancreas. Many studies have reported that Piezo1 acts as the upstream of TRPV4 by binding through phospholipase and triggers a consistent Ca^2+^ influx, leading to a variety of effects including adhesion junctions disruption and monocyte adhesion in endothelial cell [85], attenuated proliferation in osteoblastic cell [173], cell necrosis in pancreatic acinar cell [172]. One possible explanation for the functional interaction is that the two channels in the same cell are responsible for different degrees and ranges of mechanical stimuli: Piezo1 transduces transient and mild mechanical signals, while TRPV4 transduces sustained and severe mechanical signals [174,175]. However, we don’t mean that Piezo1 interacts with another Piezo1 in the same cell. In fact, Piezo1 itself inherently behave as independent mechanotransducers at physiological densities [176]. The mechanosensitive interaction also exists between Piezo1/2 and K2p channel [19,177], further indicating that it is appropriate to treat the mechanotransduction as a complicated cascade network rather than a single factor effect.

## 5. Conclusions and Future Studies

The mechanosensitive Piezo1 channel is widely expressed various organs and tissues of digestive system and has complicated roles in basal cell activities and immune signal transduction acting as a mechanosensitive transducer physiologically and pathologically. Generally speaking, Piezo1 promotes digestion, gastrointestinal motility, material transport, and generation of inflammatory response, and has negative roles on ingestion, iron metabolism, and intestinal barrier within gastrointestinal tract. However, there are several questions that deserve further deep investigation. First, further researches are needed to determine the exact role of Piezo1 on defecation in human rectum and anus. Second, the non-mechanosensitive role of Piezo1 induced by ssRNA and associated molecular mechanisms in the gut remain elusive and require further investigation. Third, we have to be cautious about the methodology in Piezo1 researches. Current body of research predominantly centers on the effects of agonists or knockdown of Piezo1 in vitro, and are lack of direct evidence and model information simulating a real mechanical microenvironment change in vivo (Table 2). While there is a growing body of evidence indicating the involvement of Piezo1 in the cellular effect, biological function, disease occurrence and development in digestive system, it is necessary to further explore the clinical translations of Piezo1 confirmed in specific mechanical microenvironment.

## Figures and Tables

**Figure 1 ijms-24-12953-f001:**
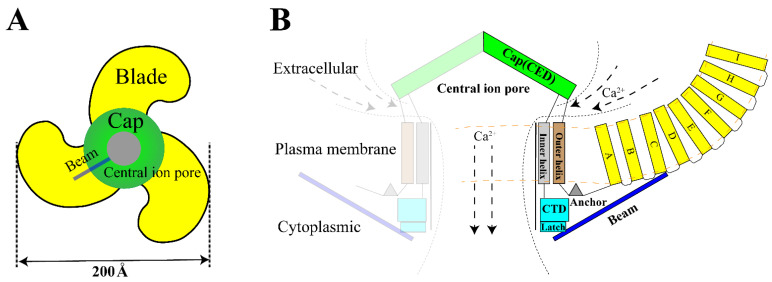
Structure of Piezo1. (**A**). The trimeric Piezo1 is a three-blade propeller with central pore from extracellular view. (**B**). Membrane view of the trimeric Piezo1. The propeller blade consists of several four-transmembrane helix bundles with the same topology called “Piezo Repeat” range A-I. CED. C-terminal extracellular domain, CTD. C-terminal domain.

**Figure 2 ijms-24-12953-f002:**
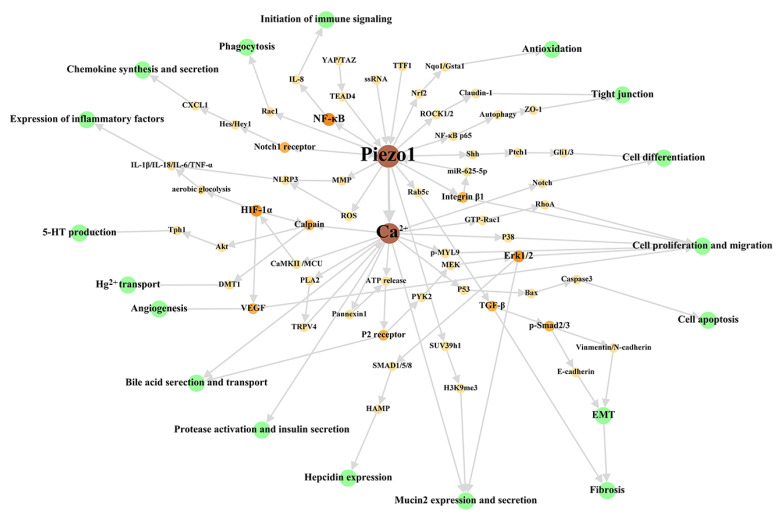
Cellular signal network of Piezo1 in digestive system. The green circles represent cellular effects. The orange part is the signal molecular between Piezo1/Ca^2+^ part and cellular effects part with gradually darkened color and increased circular diameter according to output. We dissociate the downstream Ca^2+^ signal pathway in studies stated clearly that Piezo1 functions through Ca^2+^ influx to make this figure scientific enough, although the open of Piezo1 mainly results in Ca^2+^ influx as general believed. The references associated with every signal pathway can be reviewed in Table 2. ROS: reactive oxygen species; MMP: Mitochondrial membrane potential; MCU: mitochondrial calcium uniporter; EMT: epithelial-mesenchymal transition.

**Figure 3 ijms-24-12953-f003:**
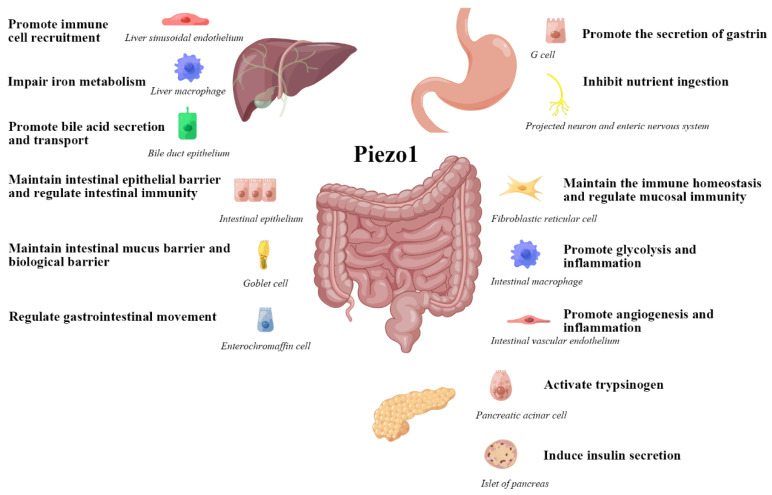
The role of Piezo1 in the biological function of the digestive system. This figure is drawn with the help of an opening website(www.figdraw.com). Export ID: WUTRY4004e, accessed on 23 January 2023.

**Figure 4 ijms-24-12953-f004:**
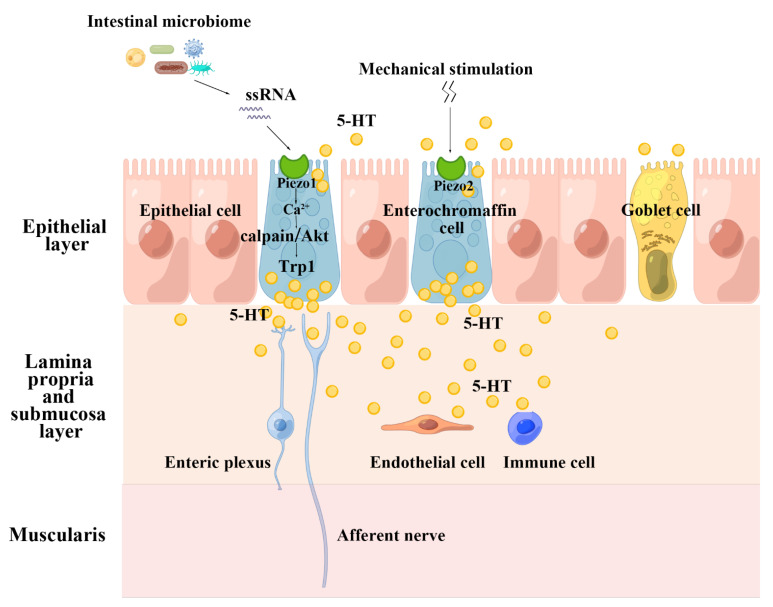
The pattern of Piezo promotes intestinal 5-HT synthesis. This figure is drawn with the help of an opening website(www.figdraw.com). Export ID: WRTPA2d426, accessed on 23 January 2023. Piezo protein mediates 5-HT synthesis, and 5-HT plays roles in gastrointestinal tract in intestinal motility, epithelial secretion, inflammation, vasodilation, trophic factor, and etc. ssRNA: single-stranded RNA; Trp1: tyrosinase-related protein 1; 5-HT: 5-hydroxytryptamine.

**Table 2 ijms-24-12953-t002:** Cellular effects of Piezo1 in digestive system.

Region	Distribution	Cellular Effect	Species	Intervention *	Study
Oral cavity	odontoblast cell	generate dentinal sensitivity, suppress dentinogenesis, conduct sensory	human	probe, micropipettes, fluid shear stress, shRNA	[40,41]
	trigeminal ganglion neuron	transduct acute pain perception	human, rat	n.a.	[42]
	dental pulp stem cell	stimulate stem cell proliferation and migration	human, rat	LIPUS, siRNA	[43,44]
	squamous carcinoma cell	promote cell growth and proliferation	human	siRNA, shRNA	[45]
	acinar cell and duct-forming regions	modulate early differentiation	mouse	siRNA	[46]
Pharyngeal	stratified squamous epithelial cell	n.a.	human	n.a.	[47]
	pharyngeal muscle, pharyngeal gland, sensory neuron	regulate pharyngeal pumping and defecation	nematode Caenorhabditis elegans	RNAi	[48]
Esophagus	squamous carcinoma cell	regulate cell apoptosis, migration, and invasion	human	shRNA	[49]
Stomach	G cell	stimulate gastrin secretion	mouse	n.a.	[50]
	submucosal and myenteric plexus cell	n.a.	human, guinea pig, mouse	intraganglionic injections	[51]
	gastric cancer cell	promote cell proliferation, migration, invasion; suppress cell apoptosis; maintain cellular morphology	human, mouse	siRNA, in vivo xenograft	[52,53]
Small intestine	enterochromaffin cell	mediate 5-HT synthesis	mouse, rat	cyclic stretching, siRNA, sgRNA	[54]
	epithelial cell	activate NLRP3 inflammasome and initiate immune gene expression	human	beads, siRNA, gRNA	[55,56]
	intestinal stem cell	trigger stem-cell proliferation and differentiation	Drosophila	microfluidic chip, gRNA	[57]
	fibroblast reticular cell	promote lymphocyte recruitment, initiate mucosal antibody responses	mice	n.a.	[58]
	submucosal plexus and myenteric plexus	n.a.	human, mouse, guinea pig	intraganglionic injection	[51]
Large intestine	goblet cell	promote mucin2 expression and mucus secretion	human, mouse	hydrostatic pressure, mechanical traction, shear force, siRNA	[59,60]
	epithelial cell	activate cell autophagy, regulate expression of tight junction protein, promote Hg^2+^ transport	human	fluid shear stress, cyclic strain, shRNA, sgRNA	[61,62,63]
	adenocarcinoma cell	promote cell migration and metastasis, mediate apoptosis	human	siRNA	[64]
	macrophage	promote aerobic glycolysis and secretion of IL-6, TNF-α, IL-1β	mouse	static pressure, cyclic hydrostatic pressure, lps	[65]
	microvascular endothelial cell	promote cell migration, organization and alignment	human, mouse	shear stress	[66]
Liver	hepatocyte	reduce mitochondrial ROS, mediate cell apoptosis/necrosis, regulate expression of hepcidin	human, mouse	siRNA, pLVX-EF1α-IRES-ZsGreen1-PIEZO1 mutant constructs	[67,68]
	hepatocellular carcinoma	promote cell proliferation, migration, invasion, EMT and angiogenesis	human, mouse, rat	matrix stiffness, shRNA, in vivo xenograft	[69,70]
	hepatoblastoma	promote cell proliferation and migration	human	siRNA	[71]
	macrophage	enhance phagocytosis, regulate expression of hepcidin	mouse	membrane stretch	[72]
	hepatic sinus endothelial cell	promote CXCL1 generation and secretion	mouse	cyclic stretch	[73]
Biliary tract	bile canaliculi	promote the contraction of peritubular actin cortex	rat	n.a.	[74]
	cholangiocyte	trigger ATP secretion	mouse	osmotic pressure, siRNA	[75]
Pancreas	pancreatic acinar cell	trigger intracellular trypsin activation and cell necrosis	mouse	pancreatic duct injection	[76]
	islet β cell	induce insulin secretion	mouse, rat	circular shear stress, hypotonicity, siRNA	[77]
	pancreatic stellate cell	promote cell migration, mediate fibrogenic responses and loss of perinuclear fat droplets	human, mouse	glass pipette, fluid shear stress, spheroid traction, acidification	[78,79]

* Intervention is not involved in reagent such as Yoda1, Jedi1/2, GsMTx4, Dooku1, RR, Gd^3+^ because of they are not specific enough and do not imitate mechanical microenvironment cell exposed. n.a. not applicable.

## Data Availability

Not applicable.

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
