# Peer review of "Piezo1 in Digestive System Function and Dysfunction"

_ijms, 2023, doi:10.3390/ijms241612953_

Round 1

Reviewer 1 Report

The authors have undertaken a fairly comprehensive review of the insights on cellular effect of Piezo1 across the entire digestive system.  Based on this, new potential lines of development are discussed.  I did not have any major concerns, only several minor issues listed below:

Page 1, lines 4, “Role of declined electrogenic Na+/HCO3 co transporter NBCe1 in mucus barrier impairment and colonic inflammation”, NBCe1 is out of focus.

Abstract in Page 3, lines 7, “The goal of this comprehensive review is to provide a solid foundation for future research into Piezo1's involvement in digestive physiology and pathology.”, Please revise “Piezo1's”.

Page 3, lines 27, “Piezo1(Fam38a), first identified in 2010 by Coste B, plays roles in various physiological processed, including vascular development, red blood cell volume homeostasis, bone and epithelial homeostasis, ne ural stem cell differentiation [10,11].” Please revise this sentence.

Footnote of Table 2 in Page 8, “Gd3+” should be present with superscript.

Page 11, line 35, “Ca2+/CaMKII”.

Page 14, line 31, “DSS-induced colitis”, please define “DSS”.

Page 19, line 7, please revise “Piezo1’s”.

Moderate editing of English language required.

Author Response

Response to Reviewer 1 Comments

Thank you for the careful censor! Response to your comments as following:

Point 1: Page 1, lines 4, “Role of declined electrogenic Na+/HCO3 co transporter NBCe1 in mucus barrier impairment and colonic inflammation”, NBCe1 is out of focus.

Response 1: Thanks to your carefully censor and you are right! We are sorry for this mistake. This sentence is a running title for another paper independent of this manuscript. It is my mistake to use the format of that title page without delete running title. I have deleted this sentence in Page 1 of the revised manuscript (Please see the attachment). The paper has been carefully revised again by Professor Zhang Lei to improve the grammar and coherence according to your valuable comments.

Point 2: Abstract in Page 3, lines 7, “The goal of this comprehensive review is to provide a solid foundation for future research into Piezo1's involvement in digestive physiology and pathology.”, Please revise “Piezo1's”.

Response 2: Thank you for this thoughtful comment. We have modified this sentence in Page 3, lines 7 (Please see the attachment) of the revised manuscript as following: “The goal of this comprehensive review is to provide a solid foundation for future researches about Piezo1 in digestive system physiologically and pathologically.”.

Point 3: Page 3, lines 27, “Piezo1(Fam38a), first identified in 2010 by Coste B, plays roles in various physiological processed, including vascular development, red blood cell volume homeostasis, bone and epithelial homeostasis, ne ural stem cell differentiation [10,11].” Please revise this sentence.

Response 3: We regret there were problems with the English. We have modified this sentence in Page 4, lines 1 of the revised manuscript (Please see the attachment) as following: “First identified in 2010 by Coste B, Piezo1 (Fam38a) plays important roles in maintaining various cellular effects such as bone and epithelial homeostasis, neural stem cell differentiation, macrophage polarization, and regulating biological functions including vascular development, red blood cell volume homeostasis, inflammation response generation and etc”.

Point 4: Footnote of Table 2 in Page 8, “Gd3+” should be present with superscript.

Response 4: Thank you for this thoughtful feedback. We have corrected this error as “Gd3+” listed in Page 9, lines2 (Please see the attachment).

Point 5: Page 11, line 35, “Ca2+/CaMKII”.

Response 5: Thank you for this thoughtful feedback. We have corrected this error in Page 11, lines 41 of the revised manuscript (Please see the attachment) as following: “This antioxidative response may be linked to the increase in Ca2+/CaMKII seen under stress conditions after TRPV1 channel activation”.

Point 6: Page 14, line 31, “DSS-induced colitis”, please define “DSS”.

Response 6: We appreciate this kind recommendation. DSS is the abbreviation of “Dextran Sulfate Sodium Salt”, which is widely used to induce colitis in animal models. We have rewritten this sentence in Page 14, lines 37 of the revised manuscript (Please see the attachment) as the following: “The activated Piezo1 promotes the shift of macrophages metabolism into aerobic glycolysis and increases the secretion of IL-1β, IL-6, TNF-α, while knock-out of Piezo1 inhibits aerobic glycolysis in colon macrophages of colitis induced by dextran sulfate sodium salt”.

Point 7: Page 19, line 7, please revise “Piezo1’s”.

Response 7: We regret there were problems with the English. We have modified this sentence in Page 19, line 31 of the revised manuscript as follows: “While there is a growing body of evidence indicating the involvement of Piezo1 in the cellular effect, ……”. Thank you again for your helpful censor!

Reviewer 2 Report

The paper is interesting and well written. The strucutre is well defined, the methodology is adequate and coerent with endpoints. Discussion and references are well defined. The paper falls in the aim of the journal and it is acceptable for publication.

Minor english editing

Author Response

Response to Reviewer 2 Comments

Thank you for the careful censor! Response to your comments as following:

Point 1: The paper is interesting and well written. The structure is well defined, the methodology is adequate and coherent with endpoints. Discussion and references are well defined. The paper falls in the aim of the journal and it is acceptable for publication.

Response 1: We appreciate the reviewer for this kind recommendation. And this paper has been carefully revised again by Professor Zhang Lei to improve the grammar and coherence considering your comments on the quality of English (Please see the attachment).

Thank you again for your helpful censor!

Reviewer 3 Report

The review of Jing He et al. provides insights on role of Piezo1 in various aspects of functioning f digestive system including ingestion and digestion, material metabolism, enteric nervous system, intestinal barrier, and inflammatory response etc.

Here are some major and some minor problems to be fixed:

1. The title is incomplete. Maybe the role of... in the functions of digestive system in health and disease, or in physiology of GIT or similar.

2. Please provide full name before using abbreviation for the first time: section 1, paragraph 1 TRPV[3] , Piezo1/2[4] , K2p [5] , BKCa [6]

3. Conclusions are lacking. Future research is not enough and does not compensate for lack of conclusions.

4. Even if very briefly, it is worth mentioning some of extra-GIT fnctions of Piezo-1 receptors/channels for general overview of the  biological role. 

no specific comments

Author Response

Response to Reviewer 3 Comments

Thank you for the careful censor! Response to your comments as following:

Point 1: The title is incomplete. Maybe the role of... in the functions of digestive system in health and disease, or in physiology of GIT or similar.

Response 1: We appreciate this kind recommendation. We have revised this title as follows depending on your valuable suggestion: “Role of mechanosensitive Piezo1 in the functions of digestive system in health and disease”. The paper has been carefully revised again by Professor Zhang Lei to improve the grammar and coherence according to your valuable comments (Please see the attachment).

Point 2: Please provide full name before using abbreviation for the first time: section 1, paragraph 1 TRPV[3] , Piezo1/2[4] , K2p [5] , BKCa [6]

Response 1: Thank you for this thoughtful comment. We have rewritten this sentence in Page 3, lines 22 of the revised manuscript (Please see the attachment) as the following: “The mechanosensitive ion channels in digestive system include transient receptor potential vanilloid family (TRPV), Piezo1/2, two pore-domain potassium channels (K2p), large-conductance Ca2+-activated potassium channel (BKCa) and others.”.

Point 3: Conclusions are lacking. Future research is not enough and does not compensate for lack of conclusions.

Response 1: We appreciate this kind recommendation. We have rewritten “Future studies” section in Page 19, lines 16 of the revised manuscript (Please see the attachment) as the following:

“5. Conclusion and future studies

The mechanosensitive Piezo1 channel is widely expressed various organs and tissues of digestive system and has complicated roles in basal cell activities and immune signal transduction acting as a mechanosensitive transducer physiologically and pathologically. Generally speaking, Piezo1 promotes digestion, gastrointestinal motility, material transport, and generation of inflammatory response, and has negative roles on ingestion, iron metabolism, and intestinal barrier within gastrointestinal tract. However, there are several questions that deserve further deep investigation. First, further researches are needed to determine the exact role of Piezo1 on defecation in human rectum and anus. Second, the non-mechanosensitive role of Piezo1 induced by ssRNA and associated molecular mechanisms in the gut remain elusive and require further investigation. Third, we have to be cautious about the methodology in Piezo1 researches. Current body of research predominantly centers on the effects of agonists or knockdown of Piezo1 in vitro, and are lack of direct evidence and model information simulating a real mechanical microenvironment change in vivo (Table 2). While there is a growing body of evidence indicating the involvement of Piezo1 in the cellular effect, biological function, disease occurrence and development in digestive system, it is necessary to further explore the clinical translations of Piezo1 confirmed in specific mechanical microenvironment.”

Point 4: Even if very briefly, it is worth mentioning some of extra-GIT functions of Piezo-1 receptors/channels for general overview of the biological role.

Response 1: Thank you for this helpful censor. We have added a sentence to the Introduction section (Page 4, lines 1) (Please see the attachment) to clarify briefly some extra-GIT roles of Piezo1 channel as follows:

Piezo1 is widely expressed in multiple cell types, whereas Piezo2 is believed to be predominantly expressed in neurons and intestinal enterochromaffin cells.

First identified in 2010 by Coste B, Piezo1 (Fam38a) plays important roles in maintaining various cellular effects such as bone and epithelial homeostasis, neural stem cell differentiation, macrophage polarization, and regulating biological functions including vascular development, red blood cell volume homeostasis, inflammation response generation and etc. More recently, Piezo1 channel has also been identified to transduce itch in sensory neuron which is associated with Piezo2 channel as generally believed. Therefore, Piezo1 participates in life activity widely and deeply. And more and more evidences have demonstrated the predominant and special contributions of the Piezo1 channel in the digestive system at present.”.

Thank you again for your helpful censor!

Round 2

Reviewer 3 Report

no comments

no comments

Author Response

thanks to your kind censor!